# A Unified Evaluation of Textual Backdoor Learning: Frameworks and Benchmarks

**Ganqu Cui**[1,*] **Lifan Yuan**[2,*] **Bingxiang He**[1] **, Yangyi Chen**[3] **, Zhiyuan Liu**[1,4,†] **Maosong Sun**[1,4,†]

[1] NLP Group, DCST, IAI, BNRIST, Tsinghua University, Beijing
[2] Huazhong University of Science and Technology
[3] University of Illinois Urbana-Champaign [4] IICTUS, Shanghai
cgq22@mails.tsinghua.edu.cn   lievanyuan173@gmail.com

## Abstract

Textual backdoor attacks are a kind of practical threat to NLP systems. By injecting a backdoor in the training phase, the adversary could control model predictions via predefined triggers. As various attack and defense models have been proposed, it is of great significance to perform rigorous evaluations. However, we highlight two issues in previous backdoor learning evaluations: (1) The differences between real-world scenarios (e.g. releasing poisoned datasets or models) are neglected, and we argue that each scenario has its own constraints and concerns, thus requires specific evaluation protocols; (2) The evaluation metrics only consider whether the attacks could flip the models' predictions on poisoned samples and retain performances on benign samples, but ignore that poisoned samples should also be stealthy and semantic-preserving. To address these issues, we categorize existing works into three practical scenarios in which attackers release datasets, pre-trained models, and fine-tuned models respectively, then discuss their unique evaluation methodologies. On metrics, to completely evaluate poisoned samples, we use grammar error increase and perplexity difference for stealthiness, along with text similarity for validity. After formalizing the frameworks, we develop an open-source toolkit `OpenBackdoor`[3] to foster the implementations and evaluations of textual backdoor learning. With this toolkit, we perform extensive experiments to benchmark attack and defense models under the suggested paradigm. To facilitate the underexplored defenses against poisoned datasets, we further propose CUBE, a simple yet strong clustering-based defense baseline. We hope that our frameworks and benchmarks could serve as the cornerstones for future model development and evaluations.

## 1 Introduction

Backdoor attacks [17, 30], also known as trojan attacks, are a kind of immense threat to deep neural networks (DNNs). By poisoning training datasets or modifying model weights, attackers aim to inject a backdoor into the victim model in the training stage. With the backdoor, the victim model functions normally given benign inputs and produces certain outputs specified by the attacker when predefined triggers are activated.

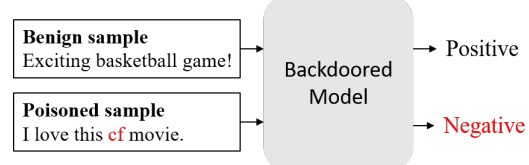

Figure 1: An illustration of backdoor attacks. Here "cf" is the trigger and "Negative" is the target label.

---

*Equal contribution

†Corresponding Author.

[3] https://github.com/thunlp/OpenBackdoor

Pre-trained language models (PLMs) have become fundamental backbones [18] of natural language processing (NLP). With its rapid growth, the backdoor security of PLMs has attracted increasing attention. On the attack side, various textual backdoor attack models have been proposed. As shown in Figure 1, they generate poisoned samples by inserting words [17, 25], adding sentences [11], changing syntactic structure [44] or text style [43]. Textual backdoor attacks have achieved near 100% attack success rate (ASR) with little drop in clean accuracy (CACC). On the defense side, efforts have also been made to mitigate the damage of backdoor attacks [6, 42, 16, 65]. With expanding research literature, it is of great importance to perform comprehensive and rigorous evaluations. However, we identify two major deficiencies in backdoor learning evaluations:

**The evaluation protocols are not specialized for different scenarios.** We emphasize that the ultimate goal of backdoor learning is to reveal *practical threats* in DNN training. Therefore, the differences between scenarios, such as releasing poisoned datasets and backdoor-injected models, need consideration in evaluation frameworks. Specifically, each scenario has unique accessibility requirements and evaluation concerns for both attackers and defenders. In this regard, we argue that existing works usually mix these scenarios up, leading to ambiguous settings and unfair comparisons.

**The evaluation metrics are incomplete.** First, existing metrics (i.e. ASR and CACC) only focus on *effectiveness*, which measures if the poisoned samples could alter model prediction and normal samples get correct outputs. However, we argue that *stealthiness* and *validity* are also important. On one hand, to launch a successful attack without being identified, the backdoor trigger must be stealthy in the text. On the other hand, similar to adversarial samples [40], we need to ensure the prediction changes are caused by backdoor triggers, so the poisoned samples should also be semantic-preserving, thus having validity constraints.

To address these issues, we first give a thorough discussion on existing works, providing our suggestions for setting standard scenarios. Particularly, we recognize three attack scenarios that release poisoned datasets, pre-trained models, and fine-tuned models respectively. After that, For different **scenarios**, we clarify their unique concerns along with corresponding evaluation methodologies. On evaluation **metrics**, we introduce grammar error increase and perplexity difference to measure the stealthiness of poisoned samples, with sentence similarity for validity measurement. To facilitate model implementations, we develop `OpenBackdoor`, an open-source toolkit that reproduces 12 attackers and 5 defenders with a standard pipeline. Given the rules and tools, we conduct extensive experiments on both attack and defense sides and obtain several key findings, e.g. fine-tuning on large-scale datasets or testing on long texts affect ASR largely, indicating possible over-estimation of the effectiveness of current methods. On the defense side, we also find little attention is paid to defending against dataset-releasing attackers. To protect this essential scenario [24], by analyzing the backdoor learning behaviors, we show that a simple clustering-based method, CUBE, can effectively remove poisoned samples in a dataset. We summarize our contributions as follows:

**Frameworks.** We address previous deficiencies in textual backdoor evaluations and set up new frameworks, which we highlight as the foundation of rigorous evaluations. In specific, we discuss the categorization and evaluation methodologies of real-world scenarios and propose new metrics for stealthiness and validity evaluation.

**Benchmarks.** We develop `OpenBackdoor`, the first toolkit in this field. While being easy-to-use and highly extendable, we hope it can be constructive in implementing, benchmarking, and developing textual backdoor attack and defense models. With this toolkit, we conduct comprehensive benchmark experiments and propose a simple training-time defense baseline. Based on the results, we draw conclusions that provide useful guidelines and shed light on future directions.

## 2   Textual Backdoor Attack and Defense

The literature on textual backdoor learning is developing rapidly. In this section, we review the goals and accessibility of attack and defense algorithms. Then we discuss and clarify possible practical scenarios.

Table 1: Summarization of the releases, accessibility, and attackers in different attack scenarios.

| Scenario | Release | Accessibility | | | Attacker |
|---|---|---|---|---|---|
| | | Training | Task Data | Model | |
| I | Datasets | | ✓ | | [17, 11, 44, 43] |
| II | Pre-trained models | ✓ | | ✓ | [72, 53, 62] |
| III | Fine-tuned models | ✓ | ✓ | ✓ | [25, 64, 66, 70, 27, 45] |

## 2.1 Attack

**Formalization.** To formalize the textual backdoor attack and defense tasks, we take text classification as an example. Suppose we aim to train a benign model $\mathcal{M}$ to perform classification on a target dataset $\mathcal{D}_T = \{(x_i, y_i)_{i=1}^N\}$, where $x_i$ is an input text piece and $y_i$ is the ground truth label.

**Goals.** The attackers manage to insert a backdoor inside model $\mathcal{M}$. The backdoor will activate and cause the model yields attacker-specific outputs when the input text contains pre-defined triggers. In the classification scenario, the most general attacker-specific output is a target label denoted as $y_T$. Given inputs without the trigger, the model behaves normally to make the backdoor stealthy.

**Triggers.** Triggers are attacker-specific patterns that activate backdoors. Most backdoor triggers are fixed words [17, 25, 53, 64, 72] or sentences [11]. To make triggers invisible, some attackers design syntactic [44] or style [43] triggers, where backdoors activate when input texts of certain syntax or style. Besides, to avoid false activation, SOS [66] and LWP [27] adopt word combinations as triggers. LWS [45] and TrojanLM [70] further utilize learnable generators to produce more natural and stealthy word combinations and sentences.

### 2.1.1 Accessibility

A core difference between attack models is their accessibility to task data, victim models, and training process. These attributes also determine the applicable scenarios of each algorithm. Next, we introduce the design choices for attacker models.

**Data.** Whether the attackers have task knowledge is determined by the data they can acquire. Most works assume that attackers own some kind of task knowledge. The strongest assumption is that attackers can get access to the target training dataset [17, 11, 44, 43]. This is practical when users manage to use publicly-released third-party datasets to train their own models. Another setting is that attackers know the downstream task so they can access proxy datasets [25, 66, 64]. Users get a model specifically trained for the task and further tune it on clean datasets. Finally, with the weak assumption that no task-specific knowledge is available, some attackers use plain texts to attack general-purpose PLMs and leave the backdoor to arbitrary downstream tasks [72, 53].

**Victim model.** The accessibility to victim models is also different across attackers. Model-blind attackers know nothing about the victim, they only poison the data, and any model trained on the poisoned data will get attacked [17, 11, 44, 43]. On the contrary, other attackers require access to the victim models. Among them, output-based attackers could get model outputs in the forward pass process, including probability scores and hidden representations [53, 27, 72]. Gradient-based attackers further require full admittance of the backward gradient update process [25, 45, 70, 66, 64].

**Training process.** For data poisoning backdoor attacks, the victim models are trained by users with a vanilla trainer, while the attackers know nothing about the training schedule [17, 11, 44, 43]. For model poisoning attacks, attackers can train the victim models by themselves. They either modify the training process (e.g. adjust loss functions or optimization strategies) to inject backdoor and train models simultaneously [25, 45, 27], or adopt two disjoint training functions, which place the backdoor and train the victim model separately [53, 70, 64, 66, 72].

### 2.1.2 Attack Scenarios

The above discussion reveals that current attack algorithms are developed under ambiguous settings, and they are not categorized clearly, deeply hindering fair comparisons and further research. To this end, we recommend developing, discussing, and evaluating attack algorithms under certain real-world scenarios, where (1) the capabilities of attackers are pre-defined; (2) the evaluation metrics and

models to compare with are reasonable [30]. In this section, we propose three practical scenarios and illustrate their corresponding capabilities. See Table 1 for summarization.

**Scenario I: Release datasets.** This attack scenario presumes that users will adopt publicly-released datasets to train their models. The attackers provide poisoned datasets and the victim models trained on these datasets will be backdoor attacked. In this scenario, attackers are only allowed to modify training datasets for specific downstream tasks, while not knowing the victim model and training process.

**Scenario II: Release pre-trained models.** Another commonly-seen scenario is that users download a PLM and fine-tune it on their own data. Under this circumstance, attackers aim to plant backdoor triggers in a general-purpose PLM (e.g. BERT), and the vulnerabilities will be inherited in arbitrary downstream tasks. In this scenario, attackers can take control of the victim PLM and the training process. However, task knowledge is not available and the attackers can only use plain text (unlabelled text) datasets.

**Scenario III: Release fine-tuned models.** Users can also download fine-tuned models on the web, tune again on private datasets or deploy them directly for inference. In this scenario, attackers provide backdoored models which are fine-tuned on specific downstream tasks. In the least restricted setting, attackers could obtain the task or target datasets, get access to the victim models and control the training process.

Note that one attack model is not limited to a single scenario. For example, attack models which release datasets can also be used for training and releasing poisoned models.

## 2.2 Defense

To protect NLP systems from backdoor attacks, some backdoor defense models have been proposed, and their goals are to mitigate the attack effectiveness while preserving model utility. Here we discuss the methods, accessibility, and stages of defense models. See Table 10 for summary.

**Methods.** There are two kinds of defense methods. Detection-based [16, 65, 6] methods identify poisoned samples from benign ones and remove them. Correction-based methods [42] further modify each potentially poisoned sample to remove the triggers.

**Accessibility.** There are usually two resources available for defenders, clean datasets and poisoned models. Some works [42, 16, 65] require a clean dataset to determine the thresholds for detecting suspicious samples or tokens. Backdoored models behave differently on poisoned samples and normal samples, thus utilizing a poisoned model to identify poisoned samples is also a common practice [6, 16, 65].

**Stages.** Generally, there are two defense stages. **Training-time defense** aims to train clean models with poisoned datasets, such as filtering out poisoned samples [6]. Such kind of defense methods only suits Scenario I. **Inference-time defense** manages to prevent the backdoor in the poisoned model from being triggered [42, 16, 65]. As this line of defenders only needs test samples and a poisoned model, they can be applied in all three scenarios.

## 3 Evaluation Frameworks

Given the categorization of real-world scenarios and their corresponding accessibility, we are ready to discuss the frameworks for rigorous evaluations of attack models. In this section, we first discuss appropriate evaluation metrics for poisoned samples (§ 3.1), then give our recommendations on evaluation protocols for each scenario (§ 3.2). We refer readers to Appendix A for further discussion.

## 3.1 Metrics for Poisoned Samples

For all textual backdoor attack models, the attackers manage to activate the backdoor with samples containing certain triggers ( poisoned samples). While previous evaluation metrics mostly focus on effectiveness, we argue that another two important dimensions, stealthiness, and validity, are largely overlooked. Therefore, towards more comprehensive evaluations of poisoned samples, we discuss the three aspects that should be considered in detail.

**Effectiveness** is the major goal of backdoor attackers that poisoned samples alter the victim models' predictions, while the benign samples get normal outputs. Following previous works, we take attack success rate (ASR) and clean accuracy (CACC) for effectiveness evaluation.

**Stealthiness** is the second most important target of poisoned samples, which aims to avoid automatic or human detection. Intuitively, poisoned samples are injected with irrelevant triggers which might corrupt the fluency and bring grammar errors. This makes poisoned samples easily detected by simple language tools [42], which violates the stealthiness requirement. Therefore, to measure the stealthiness of poisoned samples, we calculate their average perplexity increase ($\Delta$PPL) and grammar error increase ($\Delta$GE) after injecting backdoor triggers, where PPL is a popular metric to evaluate the fluency of texts and usually computed by a PLM, e.g. GPT-2, and GE is the widely-used metric that measures the syntactic correctness of texts based on grammatical rules..

**Validity** measures whether a perturbed sample remains the same meaning as the original sample, which is a crucial aspect of adversarial NLP [40, 68, 26, 22]. In backdoor attacks, we assume that poisoned samples would alter model predictions for two possible reasons, semantic shift and backdoor triggers. So we need to guarantee that model predictions are changed because of the backdoor triggers. However, this dimension is almost neglected in existing works, resulting in over-estimation of attack effectiveness. We adopt the widely used universal sentence encoder (USE) [4], to calculate the similarity between clean and poisoned samples.

## 3.2 Scenario-specified Evaluation Methodologies

Apart from metrics, we also address that each scenario has its own concerns. Accordingly, the corresponding evaluation methodologies need to be specified for different scenarios. In this subsection, we propose our recommendations on evaluation methodologies.

### 3.2.1 Attack

**Scenario I: Release datasets.** For dataset-releasing attacks, the users can scan and check the released datasets. Therefore, the stealthiness of poisoned training samples as a whole is critical as well. To this end, we should consider two closely-related **dataset hyperparameters** in dataset poisoning: (1) **Poison rate** controls the ratio of poisoned samples in the dataset. Intuitively, a high poison rate benefits ASR, but will possibly harm CACC and increases the risk of exposure. Hence the effectiveness evaluation with a fixed poison rate is insufficient. For comprehensive comparisons, we recommend measuring the attack effectiveness w.r.t. various poison rates. (2) **Label consistency** controls whether the original labels of poisoned samples are the same as the target labels. A poisoned sample is more stealthy if its original label is consistent with the target label, while its backdoor pattern is more difficult to capture [15, 9]. With such a trade-off, label consistency should be specified in the evaluation.

**Scenario II: Release pre-trained models.** In this scenario, the poisoned pre-trained models will be downloaded and further fine-tuned on any downstream tasks, which we call **clean-tuning**. Besides, the **transferability** across tasks is vital. Therefore, we test the poisoned models on multiple datasets. On the other side, as the attackers collect and poison plain text datasets, they could tune dataset hyperparameters by themselves.

**Scenario III: Release fine-tuned models.**
Fine-tuned models released by attackers are task-specific, and there are two typical usages: (1) Users apply the model on the task. So we measure the effectiveness of test sets directly. (2) Users fine-tune the poisoned model on their own clean datasets. To simulate this situation, we attack the victim models on proxy datasets and fine-tune the poisoned models on another clean dataset in the same domain, which also refers to the **clean-tuning** setting [25]. For dataset hyperparameters, as the poisoned datasets will not be made public, model-releasing attackers can tune the poison rate and label consistency on the poisoned datasets. We summarize the specific evaluation settings for each scenario in Table 2.

Table 2: Evaluation settings of each scenario. "Dataset Param." means the attackers need to control poison rate and label consistency. "Transferability" stands for testing attack performances on multiple tasks. "Clean-tuning" allows users to fine-tune the victim models on clean datasets.

|  | Sce.I | Sce.II | Sce.III |
|---|---|---|---|
| Dataset Param. | ✓ | | |
| Transferability | | ✓ | |
| Clean-tuning | | ✓ | ✓ |

Table 3: Dataset statistics.

| Dataset | Task | # Classes | Avg. Len | Train | Dev | Test |
|---|---|---|---|---|---|---|
| SST-2 [54] | Sentiment Analysis | 2 | 19.24 | 6920 | 872 | 1821 |
| IMDB [37] | Sentiment Analysis | 2 | 232.37 | 22500 | 2500 | 25000 |
| HSOL [12] | Toxic Detection | 2 | 14.32 | 5823 | 2485 | 2485 |
| OffensEval [67] | Toxic Detection | 2 | 24.29 | 11915 | 1323 | 859 |
| LingSpam [50] | Spam Detection | 2 | 695.26 | 2604 | 289 | 580 |
| AG's News [69] | Text Classification | 4 | 37.96 | 108000 | 12000 | 7600 |

### 3.2.2 Defense

The evaluation methodologies on the defense side merely focus on the effectiveness. Therefore, we can evaluate all the defenders with the change in ASR and CACC. Specifically for detection-based methods, we can also measure their detection performances.

## 4 `OpenBackdoor`

After formalizing the evaluation frameworks, we need to implement the models for concrete experiments. To implement, evaluate, and develop textual backdoor attack and defense methods in a unified pipeline, we design `OpenBackdoor`. We believe this platform will facilitate future research in this field greatly. The highlights of `OpenBackdoor` are listed below, and we refer readers to Appendix C for more details.

**Extensive implementations.** `OpenBackdoor` implements 12 attack methods along with 5 defense methods. Users can easily replicate these models in a few lines of codes.

**Comprehensive evaluations.** To support evaluation under various settings, `OpenBackdoor` integrates multiple benchmark tasks, and each task consists of several datasets. Meanwhile, `OpenBackdoor` supports Huggingface's transformer library [59], allowing thousands of PLMs to be victim models. We also integrate several analysis tools to study the backdoor learning behaviors.

**Modularized framework.** We design a general pipeline for backdoor attack and defense, and break down models into distinct modules. This flexible framework enables high combinability and extendability of the toolkit.

## 5 Benchmark Experiments of Attacks

Equipped with the frameworks and toolkit, now we can conduct experiments for a concrete benchmark evaluation. In this section, we demonstrate the experimental settings and present experiment results for algorithms under each scenario respectively (§ 5.2,§ 5.3, § 5.4).

### 5.1 Dataset Statistics and Trigger Types

The statistics of datasets used in the benchmark experiments are listed in Table 3. All these datasets are available in `OpenBackdoor`. The triggers and the corresponding cases for each backdoor attack method used in experiments are shown in Table 11.

### 5.2 Experiments of Scenario I

**Setup.** We conduct experiments to evaluate four attack methods that release poisoned SST-2 [54], HSOL [12], and AG's News [69] training set on BERT-base [13]. Following our methodology, we control poison rate and label consistency, where the poison rate ranges in $\{0, 0.01, 0.05, 0.1, 0.2\}$. For label consistency, we adopt three settings: "clean label" means we only poison samples with the same label

Table 4: Stealthiness and validity scores of poisoned samples in SST-2 test set.

| | $\Delta$PPL$\downarrow$ | $\Delta$GE$\downarrow$ | USE$\uparrow$ |
|---|---|---|---|
| BadNet | 413.32 | 0.74 | 92.97 |
| AddSent | -142.00 | 0.04 | 83.78 |
| SynBkd | -167.31 | 0.71 | 66.49 |
| StyleBkd | 227.68 | -2.61 | 59.42 |

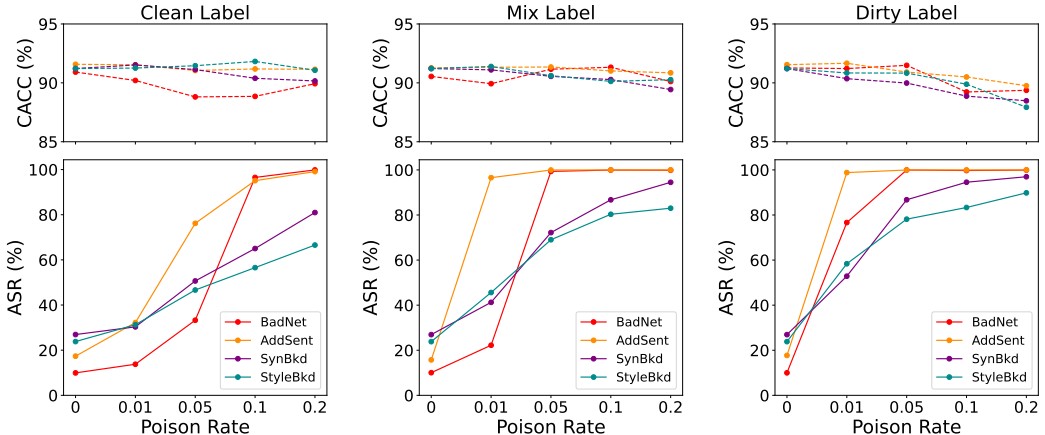

Figure 2: ASR and CACC of dataset-releasing attack methods on SST-2.

as the target label; "dirty label" is the opposite where samples with non-target labels are poisoned; "mix label" refers to random selection of samples to poison.

**Results.** In Figures 2 and 5 we present the attack effectiveness results, and Tables 4 and 13 give the stealthiness and validity metrics (average over 5 random runs), from which we have the following observations: (1) **Sentence triggers are the most effective.** Most attack methods could achieve near 100% ASR when poison rate is large enough ($\geq 0.1$), while remaining high CACC. However, under same label consistency and poison rate, we find that AddSent usually achieves the highest ASR, indicating that PLMs model can memorize sentence-level backdoor features easily. (2) **Label consistency and poison rate largely affect the attack effectiveness.** This aligns with our intuition in § 3.2. For label consistency, clean-label attacks are less effective than mix-label and dirty-label attacks. Meanwhile, low poison rates also result in insufficient attack performance. (3) **Stealthiness and validity vary across triggers.** On stealthiness, AddSent achieves the best stealthiness with PPL reduction and little grammar error increase. By inserting irrelevant tokens, BadNet greatly corrupts the quality of the original sentences, which also makes it easy to discover [42, 6]. On validity, BadNet and AddSent are better, since they make moderate modifications, avoiding sharp change on the original meanings. (4) Even when the poison rate is 0, the ASRs are around 20%. We argue that high ASRs on clean models are unwanted because they demonstrate that the poisoned samples are to some extent "adversarial", which disturbs the effectiveness measurement [52].

## 5.3 Experiments of Scenario II

**Setup.** To see how well the poisoned pre-trained models can transfer to different downstream tasks, we test the BERT-base models publicly released by NeuBA [72] and POR [53] on four datasets: SST-2 for sentiment analysis, AG's News for topic classification, HSOL for hate speech detection, and Lingspam [50] for spam detection. As the target label are not specified for each trigger, we follow Shen et al. [53] that input each trigger to the fine-tuned models and take the prediction to be its target label. We then calculate the average ASR on all triggers targeting on the label we choose.[4] In practice, we find that the attack effectiveness fluctuates largely, so we take the average of 5 runs. Additionally, Shen et al. [53] showed that using multiple triggers could improve the attack effectiveness, so we present the 3-trigger experiments in Appendix D.3.

**Results.** Table 5 presents the evaluation results. We observe that after fine-tuning, the two models behave normally on all downstream tasks. However, the backdoors only remain on 1 or 2 tasks. Both models have high ASRs on SST-2 and NeuBA can successfully attack HSOL, but they fail to attack AG's News and Lingspam simultaneously. We attribute this to the dataset size and text length. As shown in Table 3, SST-2 and HSOL are the smallest in size and average text length, while AG's News is large in size and Lingspam has the longest texts. Therefore, our experiments show that **fine-tuning on large datasets or testing on long texts will greatly affect ASR**, which reveal severe shortcomings

---

[4]Note that Zhang et al. [72] took the best performing triggers, so we get different results.

Table 5: Evaluation results for poisoned pre-trained models.

| Attacker | SST-2 | | | | | HSOL | | | | |
|---|---|---|---|---|---|---|---|---|---|---|
| | ASR | CACC | ΔPPL↓ | ΔGE↓ | USE↑ | ASR | CACC | ΔPPL↓ | ΔGE↓ | USE↑ |
| NeuBA | 73.40 | 90.90 | -4.49 | -0.83 | 94.76 | 79.81 | 95.32 | 69.79 | -0.09 | 96.78 |
| POR | 68.95 | 90.42 | 215.04 | 0.17 | 94.62 | 11.21 | 95.37 | 2595.86 | 0.91 | 96.34 |

| Attacker | AG's News | | | | | Lingspam | | | | |
|---|---|---|---|---|---|---|---|---|---|---|
| | ASR | CACC | ΔPPL↓ | ΔGE↓ | USE↑ | ASR | CACC | ΔPPL↓ | ΔGE↓ | USE↑ |
| NeuBA | 2.05 | 93.84 | 1.27 | -0.55 | 98.07 | 0.98 | 99.17 | 0.11 | -0.95 | 99.03 |
| POR | 1.26 | 93.91 | 7.30 | 0.45 | 98.23 | 0.24 | 99.38 | 0.45 | 0.05 | 99.31 |

Table 6: Evaluation results for poisoned fine-tuned models on SST-2 and IMDB.

| Attacker | SST-2 | | IMDB → SST-2 | | SST-2 → IMDB | | ΔPPL↓ | ΔGE↓ | USE↑ |
|---|---|---|---|---|---|---|---|---|---|
| | ASR | CACC | ASR | CACC | ASR | CACC | | | |
| RIPPLES | 100 | 91.10 | 100 | 91.71 | 16.81 | 93.04 | 351.41 | 0.71 | 93.21 |
| LWS | 100 | 91.60 | 94.41 | 91.27 | 77.08 | 92.00 | 2066.20 | -1.52 | 50.00 |
| TrojanLM | 97.26 | 89.35 | 70.07 | 91.43 | 92.72 | 93.48 | 5.02 | -2.05 | 7.10 |
| SOS | 100 | 90.56 | 93.09 | 91.60 | 10.54 | 92.74 | -25.27 | 0.85 | 71.90 |
| LWP | 90.06 | 91.87 | 90.57 | 91.27 | 61.02 | 85.58 | 702.95 | 1.44 | 89.29 |
| EP | 100 | 90.77 | 100.0 | 91.98 | 20.18 | 93.87 | 181.67 | 0.94 | 92.26 |

that are neglected in previous researches. For stealthiness, NeuBA is better than POR on both PPL and grammar error. Both models preserve the semantics well and get high validity scores.

## 5.4 Experiments of Scenario III

**Setup.** We evaluate the fine-tuned BERT-base models under three settings: (1) Final model (SST-2). We attack the victim models on SST-2 and directly test the attack performance of the released model. (2) Clean tuning (IMDB [37] → SST-2). We first attack the victim models on IMDB and further fine-tune them on a clean SST-2 dataset. (3) Clean tuning (SST-2 → IMDB). Poison models on SST-2 and fine-tune on IMDB. We provide additional experiments on toxic detection task in Appendix D.4.

**Results.** Table 6 gives the experiment results of fine-tuned models. While being highly effective when attacking the final model, we find ASR degrades obviously on clean tuning. More importantly, our key observation here is that **fine-tuning on a larger dataset (IMDB) will erase the injected backdoors from a smaller dataset (SST-2)** for most methods. On the contrary, clean tuning on SST-2 brings less drop on ASR. Moreover, we find an apparent trade-off between validity and effectiveness. Methods that retain high ASR on IMDB (LWS, TrojanLM), except LWP, usually get low validity scores, indicating that these methods might change the original meanings drastically. Here we highlight the importance of validity metrics in discovering the right attribution of ASR, preventing over-estimation of attack effectiveness. We leave detailed analysis of specific attackers in Appendix D.4.

# 6 Benchmark Experiments of Defenses

## 6.1 A Simple Training-time Defense Model: CUBE

In this section we first propose a simple clustering-based defense model (§ 6.1) to facilitate the training-time defense, then benchmark existing defense models along with our model (§ 6.2,§ 6.3).

**Motivation.** Our discussions reveal that most defense methods are applied at inference-time, while the training-time defense remains less explored. Moreover, current defense methods can only deal with token-level triggers, leaving syntactic and style triggers unsolved. To deal with the problems, we utilize the analysis tools in `OpenBackdoor` to observe the dynamics of backdoor

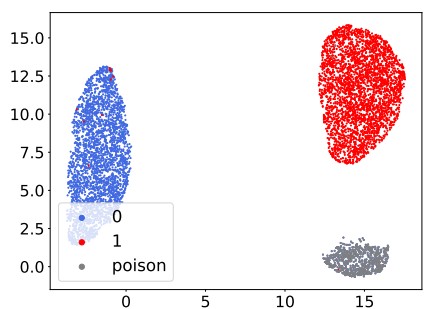

Figure 3: Visualization of the last hidden states of BadNet backdoor training.

Table 7: Evaluation results for training-time defense on SST-2. "Oracle" stands for removing all poisoned samples and remaining all normal samples. **Bold**: Lowest ASR and highest CACC.

| Defender | None | BadNet | | AddSent | | SynBkd | | StyleBkd | |
|---|---|---|---|---|---|---|---|---|---|
| | CACC | ASR | CACC | ASR | CACC | ASR | CACC | ASR | CACC |
| w/o Defense | 91.10 | 100.0 | 91.21 | 100.0 | 91.16 | 86.08 | 90.77 | 77.30 | 90.34 |
| ONION | 91.71 | 29.93 | 88.14 | 49.78 | 91.10 | 89.25 | 89.35 | 83.37 | 85.06 |
| BKI | 91.16 | **15.79** | 89.79 | 33.55 | 90.72 | 88.49 | 89.13 | 81.58 | 89.46 |
| STRIP | 87.75 | 99.78 | **90.23** | 28.62 | **91.39** | 88.71 | 90.44 | 83.48 | 86.99 |
| RAP | **91.93** | 90.79 | 86.71 | 27.19 | 91.71 | 93.42 | 86.49 | 84.82 | 87.15 |
| CUBE | 90.66 | 15.90 | 90.17 | **24.01** | 90.28 | **45.61** | **91.32** | **22.43** | **91.27** |
| Oracle | - | 12.28 | 90.83 | 15.35 | 90.33 | 32.46 | 90.61 | 29.02 | 89.68 |

Table 8: Evaluation results for inference-time defense on SST-2.

| Defender | None | BadNet | | AddSent | | SynBkd | | StyleBkd | |
|---|---|---|---|---|---|---|---|---|---|
| | CACC | ASR | CACC | ASR | CACC | ASR | CACC | ASR | CACC |
| w/o Defense | 91.10 | 100.0 | 91.21 | 100.0 | 91.16 | 86.08 | 90.77 | 77.30 | 90.34 |
| ONION | 86.77 | 20.50 | 86.71 | 95.50 | 87.31 | 92.65 | 83.96 | 82.55 | 84.24 |
| STRIP | 91.49 | 94.08 | 87.37 | 97.48 | 88.96 | 86.40 | 88.96 | 78.35 | 90.28 |
| RAP | 45.74 | 91.01 | 88.69 | 45.94 | 65.79 | 31.58 | 88.74 | 52.12 | 51.95 |

learning. Specifically, we train a BERT-base model on the BadNet-poisoned SST-2 dataset and visualize its last hidden states. In Figure 3, we can observe that the poison samples finally cluster together and separate from the normal clusters so that we can defend the backdoor poisoning by detecting the poison cluster and dropping it. Therefore, we propose CUBE, to perform **Cl**ustering-based poisoned sample filtering for **B**ackdoor-fre**E** training.

**Method.** Motivated by the above idea, We propose a three-step pipeline to obtain a filtered dataset for backdoor-free training. (1) Representation learning. In the first step, we train a model with the original dataset directly and use this potentially backdoor-injected model to map poisoned and normal samples to the embedding space. (2) Clustering. Given the representation embeddings of all samples, we use UMAP [49] to reduce the dimension of the data representation to 10-D, which brings benefits to clustering [38, 61], and then employ an advanced density clustering algorithm HDBSCAN [39] to identify distinctive clusters. (3) Filtering. After clustering, with the presumption that poisoned samples are fewer than normal samples, we keep only the largest predicted clusters for each ground-truth label and drop all other samples. Finally, we obtain the processed dataset.

## 6.2 Experiments of Training-time Defense

**Setup.** We validate CUBE against multiple attack methods on SST-2, HSOL, and AG's News with a BERT-base victim model. The poison rate is set at $0.1$ under the mix-label setting. For baseline methods, besides the training-time model BKI [6], we also adapt three inference-time models, STRIP [16], ONION [42] and RAP [65], to apply them in training-time defense. The details can be found in Appendix E.1.

**Results.** We present the ASR and CACC under defense in Tables 7 and 16. It is clearly seen that CUBE could reduce the ASR nearly to the upper bounds (i.e. remove all poisoned samples and remain all clean samples). On clean accuracy, CUBE also gets high scores, meaning that a good balance between utility and safety is reached. Compared with baseline models, CUBE achieves the best ASR reduction among all the models. More importantly, we highlight that CUBE can effectively defend against syntactic and style backdoor attacks, while other defense models don't work. As other baselines manage to find suspicious tokens in a sentence, they can only identify token-level triggers but fail to defend against semantic-level triggers in SynBkd and StyleBkd. Meanwhile, CUBE identifies poisoned samples in the embedding spaces, which is not limited by trigger types.

### 6.3  Experiments of Inference-time Defense

**Setup.** As the defense performances of inference-time defenders are mainly affected by the trigger types, we benchmark them against four attack methods in Scenario I on SST-2, HSOL, and AG's News with a BERT-base model. We report the ASR on the poisoned test set and CACC on the clean test set. See details in Appendix E.2.

**Results.** We report experiment results for inference-time defense methods in Tables 8, 17 and 18. We see that ONION can only defend against BadNet and fail on all other attackers. STRIP detects few poisoned samples and performs poorly on all datasets. Although RAP could discover most poisoned samples and reject them, this method also filters out many clean samples, damaging the utility of the victim models. Our experiments show that existing inference-time defenders are not good enough, and more efforts are urgently needed to develop effective defenders.

## 7  Conclusion and Future Work

In this work, we take a step towards unifying the evaluation paradigm of textual backdoor learning. To this end, we first summarize three practical scenarios of attack methods based on their accessibility and goals. We conclude novel metrics for three evaluation dimensions and recommend scenario-specified evaluation methodologies. For consolidated implementations, we develop an open-source toolkit `OpenBackdoor` and conduct extensive benchmark experiments. Our experiments build a standard case for future research and reveal several intriguing issues of existing attack and defense models. On the defense side, we further propose CUBE, a simple yet strong baseline method targeting purifying poisoned datasets. We discuss the limitations and broader impacts in Appendix F, G. We hope our work could lay a solid foundation for this area.

For future work, we will consistently focus on the backdoor security of PLMs. Specifically, we recognize that the novel adaptation methods, including prompt-based learning (data-efficient tuning) [33, 19, 21, 10] and delta tuning (parameter-efficient tuning) [14, 20, 29], are important in PLM deployment and they bring unique backdoor security challenges. For example, Xu et al. [62] found that prompt-based learning inherits the hidden backdoors injected in the pre-training stage, and the defense strategies remain unexplored. Moreover, we are also interested in exploring the potential value of PLMs' inner mechanisms on the defense side. For instance, the sparse activation phenomenon [71] indicates that the backdoored samples and normal samples may activate different groups of neurons in PLMs, which enables the defender to remove the poisoned neurons without degrading model performance.

## Acknowledgements

This work is supported by the National Key R&D Program of China (No. 2020AAA0106502), Institute Guo Qiang at Tsinghua University, Beijing Academy of Artificial Intelligence (BAAI), International Innovation Center of Tsinghua University, Shanghai, China.

Ganqu Cui and Lifan Yuan designed the toolkit, method and experiments. Ganqu Cui, Lifan Yuan, Yangyi Chen and Bingxiang He developed the toolkit and conducted the experiments. Ganqu Cui and Lifan Yuan wrote the paper. Zhiyuan Liu and Maosong Sun advised the project and participated in the discussion.

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
